# Tendency to Worry and Fear of Mental Health during Italy’s COVID-19 Lockdown

**DOI:** 10.3390/ijerph17165928

**Published:** 2020-08-15

**Authors:** Chiara Baiano, Isa Zappullo, Massimiliano Conson

**Affiliations:** Laboratory of Developmental Neuropsychology, Department of Psychology, University of Campania Luigi Vanvitelli, 81100 Caserta, Italy; chiara.baiano@unicampania.it (C.B.); isa.zappullo@unicampania.it (I.Z.); ambulatorio.npee@gmail.com (t.L.G.)

**Keywords:** mental health, worry, anxiety, threat, mindfulness, COVID-19

## Abstract

Background: We tested whether the tendency to worry could affect psychological responses to quarantine by capitalizing on the opportunity of having collected data before the COVID-19 outbreak on measures of worry, anxiety, and trait mindfulness in a group of university students. Methods: Twenty-five participants completed self-report measures assessing worry (Penn State Worry Questionnaire, PSWQ), anxiety (Anxiety Sensitivity Index, ASI-3), and trait mindfulness (Mindful Attention Awareness Scale, MAAS) at T0 (pre-lockdown, 4 November 2019–17 February 2020) and T1 (at the end of lockdown, 26 April–30 April 2020). We compared assessments at the two time points in the whole sample and in high and low worriers (defined at T0 by scores on PSWQ respectively above and below 1.5 SD from mean of the Italian normative sample). Outcomes: High worriers showed at T1 a significant increase of anxiety sensitivity and fear of mental health in comparison to low worriers. Moreover, in the whole sample, at T1 trait mindfulness was inversely related to worry and fear of mental health. Interpretation: A valuable approach to support individuals experiencing anxiety related to the COVID-19 outbreak could be represented by mindfulness-based interventions improving the ability to focus attention and awareness on the present moment.

## 1. Introduction

Since the COVID-19 pandemic started in Wuhan (Hubei province) at the beginning of 2020, it spread in 208 countries with a global impact on the health care and economics [1]. In Italy, the Government implemented several restrictive measures to contain the spread of the infection, such as a strong recommendation to avoid at risk behaviors, the suspension of all non-essential businesses and school and university closure in the entire nation [2]. Moreover, on March 11, the Italian Government announced the beginning of the quarantine period (phase 1), which lasted until May 4. After, the phase 2 started with the gradual easing of Italy’s lockdown, from then on it was possible for everyone to go out to do exercise in parks, to visit relatives and loved ones, and no longer go out just to do shopping or for health reasons.

The psychological impact of quarantine was previously described in Chinese university students where about 24.9% of the persons interviewed experienced anxiety related to the COVID-19 outbreak [3]. In Italy, dysfunctional personality domains, as well as negative affectivity and detachment, emotional problems and supernatural beliefs resulted to be relevant risk factors for reduced emotional well-being during the COVID-19 lockdown [4]. In a 2020 meta-analysis, the main psychological consequences of quarantine during the 2003 SARS and the 2014 Ebola outbreaks were fears of infection, frustration, and boredom. In addition, the duration of quarantine and inadequate information contributed to psychological distress [5]. Accordingly, in a study on the psychological impact of the 2003 SARS quarantine, participants reported more negative feelings such as worry, anger, guilt, nervousness, and sadness [6]. More recently, Liu [7] showed that COVID-19 related information consumption on social media during the pandemic outbreak elicited intense worry which, in turn, increased preventive behaviors such as washing hands more regularly, staying away from crowded places and wearing face masks.

Worry is defined as an attempt to engage in mental problem-solving to deal with an issue whose outcome is uncertain but likely negative [8] or can be conceived as an apprehensive expectation about real-life concerns such as health, relations, finances, work, and school [9]. Worry is the central feature of generalized anxiety disorder [10], but it is also frequently associated with depressive rumination [11] and with psychopathological conditions such as panic disorder and obsessive-compulsive disorder [12]. However, worry can also be understood as a non-clinical subjective trait [13], with high levels of trait worry being associated with negative health outcomes and somatic health complaints [14]. Furthermore, high trait worry relevantly contributes to severity of psychological responses to traumatic events and stressors [15]. Conversely, trait mindfulness, defined as the individual’s tendency to focus attention and awareness to the current experience or the present moment [16], is related to lower post-traumatic responses to stressful events [17].

In the present study, we investigated whether individual’s proneness to worry could affect anxiety responses to the quarantine by capitalizing on the unique opportunity of having collected data before the COVID-19 outbreak on psychometrically valid measures assessing worry, anxiety and trait mindfulness in a group of young adults attending University. We used a classical measure of worry, i.e., The Penn State Worry Questionnaire (PSWQ [18,19]) to distinguish between people with high worry (i.e., participants with scores on PSWQ above 1.5 SD from mean of the Italian normative sample) and people with low worry (i.e., participants with scores on PSWQ below 1.5 SD from mean of the Italian normative sample). We expected that participants with high levels of pre-existing (before lockdown) worry could display stronger anxiety symptoms during the quarantine when compared to individuals with low levels of worry, while trait mindfulness potentially acting as a protecting factor to emotional distress. Moreover, since sex differences have been reported in predisposition to worry, anxiety and emotional distress, with woman being more worried [20], anxious [21] and at higher risk of emotional distress than men [22], here we also looked at possible sex differences in the psychological responses to the quarantine.

## 2. Materials and Methods

### 2.1. Participants

For the present study, 31 university students we recruited (14 females; mean age at T0 = 23.84 ± 2.4). All students underwent the first, pre-lockdown, assessment between 4 November 2019 and 17 February 2020 (T0), but 6 out 31 of them did not provide their consent for participating to the second assessment (T1) between 26 April and 30 April 2020 (about 18–22 days before starting the Italian phase 2; post-quarantine: from 4 May 2020). Thus, our final sample was of 25 participants (10 females; mean age at T0 = 23.84 ± 2.5).

To be included in the study, each participant had to report no clinical diagnosis or past history of psychiatric, neurological or neurodevelopmental disorders, as well as no assumption of drugs or substances acting on the central nervous system.

Recruitment of the students for the T0 assessment was possible since all the participants completed a series of self-report measures (see below) as part of their research activities at the Laboratory of Developmental Neuropsychology, at the Department of Psychology, University of Campania Luigi Vanvitelli (Caserta, Italy). To achieve the T1 assessment, participants were reached again through their email addresses and required to complete the same measures as at T0 after having provided the online written informed consent.

The research protocol was approved by the local ethics committee (code: N:30/2020) and the procedures were in accordance with the ethical standards laid down in the 1964 Declaration of Helsinki. The written informed consent was provided by all participants before each of the two assessments.

### 2.2. Measures

Participants underwent three psychometrically valid self-report questionnaires assessing worry, anxiety, and trait mindfulness. Each measure was administered at both T0 and T1. The second assessment at T1 was conducted through a specific online platform (Google Forms, Google Inc., Mountain View, CA, USA).

The Penn State Worry Questionnaire (PSWQ [18,19]) assesses worry as a trait measure, as it refers to the subjective tendency of worrying regardless of the situations. The items do not relate to the content of the person’s concerns but consider the critical aspects of a significant worry, such as the intensity of the process, its excessiveness and the sense of uncontrollability. The PSWQ includes 16 items with the total score ranging from 16 to 80. Here, the raw scores were first transformed in z values [19]. Then, participants with a PSWQ z score of +1.5 SD from mean of normative sample were included in the high worry subgroup, whereas participants with a z score below 1.5 SD were included in the low worry subgroup.

The Anxiety Sensitivity Index-3 (ASI-3 [23,24]) is a self-report scale that measures the degree to which one is concerned about possible negative consequences of anxiety symptoms. The ASI-3 provides three sub-scales. The physical concerns factor refers to the fear of somatic anxiety symptoms, which are believed to lead to a catastrophic physical issue. The social concerns sub-scale is about the belief that anxiety symptoms will manifest in a social context. The cognitive concerns factor refers to the fear of the mental correlates of anxiety symptoms, considered as signals of a mental health disorder. The ASI-3 includes 18 items with a total score ranging from 0 to 72.

The Mindful Attention Awareness Scale (MAAS [16]) measures the individual differences in daily mindful states conceived as the dispositional capability to pay attention to what is occurring in the present moment. In particular, the MAAS deals with the presence of attention and awareness of what is occurring in the present rather than with mindfulness attributes such as acceptance, trust, empathy, gratitude. The scale is composed by 15-items with a total score ranging from 0 to 6.

### 2.3. Statistical Analysis

The percentage of high and low worriers at T0 and T1 in the whole group, and in males and females separately, was computed and then compared with a chi-squared test.

PSWQ, ASI-3, and MAAS scores between T0 and T1 were computed, and the two-tailed Mann–Whitney U test was used to evaluate differences between the two assessments in the whole group, and in females and males separately.

PSWQ, ASI-3, and MAAS scores at T0 and T1 were compared in high and low worriers by two-tailed Mann–Whitney U test.

Finally, Spearman’s correlations were conducted to examine possible relationships between trait mindfulness, worry, and anxiety, separately at T0 and T1.

All analyses were performed using SPSS version 25 (SPSS Inc., Chicago, IL, USA) and the α level was set at 0.05 for all analyses.

## 3. Results

In the total sample, high worriers were 60% at T0 and 68% at T1 (χ^2^ = 0.125; *p* = 0.724). The percentage of high worriers at T0 was 53.3% for the male group and 46.7% for the female group (χ^2^ = 0.067; *p* = 0.796). At T1, the percentage of high worriers was 58.8% for the male group and 41.2% for the female group (χ^2^ = 0.067; *p* = 0.796).

On PSWQ, ASI-3, and MAAS no significant differences between T0 and T1 (all *p* > 0.05) were found in the total sample. Moreover, no significant differences were found when comparing males and females at the two time points (all *p* > 0.05) (Table 1).

When considering high vs. low worry subgroups, the Mann–Whitney U test revealed in high worriers a significantly higher ASI-3 total score (U = 625; *p* = 0.037) and a significantly higher ASI-3 cognitive concern score (U = 63.5; *p* = 0.041) at T1 with respect to T0. In the low worry subgroup, instead, no significant differences were found between T0 and T1 (all *p* > 0.05) (Table 2).

Correlational analyses (Table 3) showed at T0 significant negative correlations between MAAS score and both ASI-3 total score (rho = −0.491; *p* = 0.013) and ASI-3 cognitive concern score (rho = −0.573; *p* = 0.003). At T1, correlational analyses showed significant negative correlations between the MAAS score and both PSWQ score (rho = −0.505; *p* = 0.010) and ASI-3 cognitive concern score (rho = −0.405; *p* = 0.044).

## 4. Discussion

In the present study, we took advantage of having collected data before the COVID-19 outbreak on self-report questionnaires assessing worry, anxiety, and trait mindfulness in a group of university students to investigate whether individual’s tendency to worry could have an impact on anxiety responses to the quarantine. By comparing assessments at T0 and T1 in the whole group we did not find significant changes in worry, anxiety, and trait mindfulness scores, even when a comparison between males and females was performed. Instead, the individual’s proneness to worry before the COVID-19 outbreak (T0) proved to be an important factor to distinguish two groups of individuals differing in the way in which anxiety responses changed throughout the quarantine. Indeed, participants with high trait worry at pre-lockdown showed at T1 a significant increase of ASI-3 total score and cognitive concerns with respect to the low trait worry subgroup. Thus, high worriers were more anxious (“fear of fear”; ASI-3 total score) [23] and specifically concerned about the mental correlates of anxiety symptoms considered signals of cognitive dyscontrol (e.g., “I worry that I might be going crazy”; ASI-3 cognitive concerns) [25]. Instead, no significant mean change about physical or social correlates of anxiety symptoms in either the high or low worry group was found.

As recalled above, the ASI-3 is composed of one higher-order factor (general anxiety sensitivity) and three lower-order dimensions, namely, physical concerns implying fear of somatic sensations, social concerns relating to fear of publicly anxiety symptoms that may cause social rejection or ridicule, and cognitive concerns referring to fear of psychological dyscontrol [25,26]. In particular, ASI-3 cognitive concerns seem to be weakly related to trait worry on the PSWQ in the general Italian population [27]. Instead, other studies showed that concerns about mental and cognitive control were most strongly correlated with worry symptoms and with a diagnosis of generalized anxiety disorder and panic disorder [28] as well as with depression and general distress [29,30]. Moreover, it has been suggested that persons can experience intense and persistent worry about their mental health analogously to how people worry about their physical health, and perceived threat to own mental health can be as distressing as threat to own physical health [31]. Our results showing that a high tendency to worry before the pandemic outbreak was related to increase of fear about own mental health are consistent with literature underscoring the relationship between proneness to worry and the capacity to deal with mental or cognitive responses to stressful events [28].

The COVID-19 pandemic has increased the appearance of emotional distress and psychopathology [32]. Specific groups of persons are more vulnerable than others to the psychosocial effects of pandemics, in particular, people who are at risk to contract the disease and people with pre-existing clinical, psychiatric, or medical conditions [33]. The present results suggest that trait worry could be considered a factor, among others, defining individuals who are at-risk of developing dysfunctional psychological responses to the pandemic exposure. For this reason, effort in detecting early signs of clinical mental health problems is of primary relevance to deal with the psychological crisis due to COVID-19 pandemic, especially in those at-risk populations.

We did not find significant sex differences in the psychological measures selected here, although at descriptive level we could observe that both at T0 and T1 worry and anxiety scores were higher in women than in men, consistent with data on European (Italian and Spanish) samples [27,34]. Instead, we found that at T1 trait mindfulness was inversely related to worry and fear of mental health, as higher levels of trait mindfulness were related to lower levels of worry and anxiety of psychological dyscontrol. This result might suggest that trait mindfulness could have protected people from maladaptive worrying during the COVID-19 lockdown, consistent with previous findings showing that the individual’s tendency focus attention and awareness on the present moment was related to lower post-traumatic responses to stressful events [17]. Thus, the inverse relationship between trait mindfulness and worry and anxiety could imply that mindfulness might help people to keep worry and fear under control.

These present results have to be interpreted while considering some study limitations. First, the present observation took into account a specific set of psychological variables. Second, a small sample was assessed, and p-values were not adjusted for multiplicity. Third, our findings derived from the assessment of a group of university students, and available literature suggest that university students are more vulnerable to develop clinical and subclinical anxiety and anxiety-related problems with respect to the general population [35,36,37]. Thus, caution is needed when generalizing the present findings to the broader population.

Notwithstanding these limitations, by comparing psychological measures collected when the COVID-19 pandemic was well beyond our worst fears with their changes, while approaching the end of the Italy’s lockdown allowed us to reveal that a high tendency to worry predisposed persons to develop a “fear of going crazy” when exposed to emotional distress, as during the quarantine.

## 5. Conclusions

In synthesis, high trait worriers at pre-lockdown showed at the end of the quarantine a significant increase of anxiety sensitivity and, in particular, of concerns about own mental health with respect to low worriers. The present observation does not allow the clarification of the mechanism behind such a significant increase of this kind of fear in high worriers but allows us to suggest that tailored interventions selected within the framework of cognitive behavioral therapy should be provided to support at-risk individuals. In particular, mindfulness-based interventions, as mindfulness-based stress reduction [38] and mindfulness-based cognitive therapy [39], are warranted since they favor enhancement of the ability to focus attention and awareness on the present moment. Both protocols have notable empirical support and have proved to be valuable choices for treating psychological responses to traumatic experiences [40,41,42]. For this reason, they could represent valuable treatment options for persons exposed to the traumatic experience of the COVID-19 pandemic.

## Figures and Tables

**Table 1 ijerph-17-05928-t001:** Worry, anxiety, and trait mindfulness before (T0) and at the end (T1) of the Italy COVID-19 lockdown, separately in female and male participants.

Measures	T0 (N = 25)				T1 (N = 25)			
	Females (N = 10)	Males (N = 15)	U	*p*	Females (N = 10)	Males (N = 15)	U	*p*
PSWQ score	57.2 ± 13.6	46.7 ± 15.2	43.5	0.080	58.7 ± 17.9	48 ± 12.4	44	0.091
ASI-3 total score	23.1 ± 13.9	15.6 ± 11.1	50.5	0.177	30.1 ± 17.2	19.6 ± 15.3	48	0.144
ASI-3 physical concern	8.4 ± 6.6	4.9 ± 5.9	49	0.160	10 ± 7.7	5.7 ± 6.7	45.5	0.103
ASI-3 cognitive concern	6.7 ± 5.5	3.1 ± 2.9	47.5	0.129	9.7 ± 7.9	5.3 ± 4.7	48	0.144
ASI-3 social concern	8 ± 4.9	7.5 ± 5.1	69.5	0.765	10.4 ± 6.3	8.7 ± 6.9	64	0.567
MAAS score	3.7 ± 0.8	4.1 ± 0.9	60	0.428	3.5 ± 1.1	3.9 ± 0.7	62	0.495

The values are expressed as mean ± standard deviation; N, number of participants; T0, before the lockdown; T1, at the end of the lockdown; PSWQ, Penn State Worry Questionnaire; ASI-3, Anxiety Sensitivity Index; MAAS, Mindful Attention Awareness Scale.

**Table 2 ijerph-17-05928-t002:** Worry, anxiety, and trait mindfulness before (T0) and at the end (T1) of the Italy COVID-19 lockdown, separately in high and low worriers.

Measures	High Worriers (N = 15)				Low Worriers (N = 10)			
	T0	T1	U	*p*	T0	T1	U	*p*
PSWQ score	61 ± 9.7	62.8 ± 9.1	96.5	0.512	35.7 ± 6.7	36.5 ± 6.8	47	0.853
ASI-3 total score	25.5 ± 11.5	34.6 ± 12.1	62.5	0.037 *	8.3 ± 5.1	7.6 ± 4.6	48	0.912
ASI-3 physical concern	9.2 ± 6.6	11.4 ± 6.7	89.5	0.345	2 ± 2.1	1.4 ± 2.2	37	0.353
ASI-3 cognitive concern	6.1 ± 4.9	10.6 ± 5.9	63.5	0.041 *	2.2 ± 2.3	1.7 ± 1.9	45	0.739
ASI-3 social concern	10.1 ± 4.8	12.6 ± 6.4	86.5	0.285	4.1 ± 2.1	4.5 ± 2.9	48.5	0.912
MAAS score	3.6 ± 0.9	3.5 ± 0.9	102.5	0.683	4.3 ± 0.6	4.1 ± 0.5	38.5	0.393

The values are expressed as mean ± standard deviation; N, number of participants; T0, before the lockdown; T1, at the end of the lockdown; PSWQ, Penn State Worry Questionnaire; ASI-3, Anxiety Sensitivity Index; MAAS, Mindful Attention Awareness Scale. * Significant at *p* < 0.05.

**Table 3 ijerph-17-05928-t003:** Correlations (rho and *p* values in brackets) between trait mindfulness, worry, and anxiety.

	PSWQ Score	ASI-3 Total Score	ASI-3 Physical Concern	ASI-3 Cognitive Concern	ASI-3 Social Concern
MAAS score					
T0	−0.353 (0.084)	−0.491 (0.013) *	−0.308 (0.135)	−0.573 (0.003) *	−0.273 (0.187)
T1	−0.505 (0.010) *	−0.357 (0.080)	−0.336 (0.101)	−0.405 (0.044) *	−0.049 (0.817)

T0, before the lockdown; T1, at the end of the lockdown; PSWQ, Penn State Worry Questionnaire; ASI-3, Anxiety Sensitivity Index; MAAS, Mindful Attention Awareness Scale. * Significant at *p* < 0.05.

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
