# Peer review of "Tendency to Worry and Fear of Mental Health during Italy’s COVID-19 Lockdown"

_ijerph, 2020, doi:10.3390/ijerph17165928_

Round 1
Reviewer 1 Report
This study investigates how the COVID-19 may have affected individuals on anxiety, worry and trait mindfulness. The findings have potentials to contribute to the current knowledge on coping with the current pandemic and their trait may play a role in such disruptive and distressful events. The study has apparent benefits, and yet the study design (or its presentation) could be much improved. With the extremely small same size, first of all, this study should include more descriptions on the sample and further elaborate its limitations based on them. No confounding factors were considered in the study, which may critically limit the study’s interpretation and its usage. Given such small sample size, it could be much stronger with qualitative information of the samples and other external factors (other than traits). A mixed-method approach would make this study more meaningful and informative than as is. Given worry/anxiety is intricate and often conflated with situational (state) components, I strongly recommend adding rich qualitative data to consider and inform about confounding factors for the key variables. The study could also be looked at from MAAS score high vs. low (e.g. trait worry presented in Table 2 in reverse) and its (trait mindfulness’s) relation with trait worry should be further clarified in text.
Another critical issue that authors would want to address is how this study may actually contribute to any potential practice or policy interventions. The (probably only) implication authors made was on p. 5 of 7. L. 184-6. “effort in detection early signs of clinical mental health problems is of primary relevance to deal with the psychological crisis due to COVID‐19 pandemic, especially in at-risk populations.” Since the study results are from non-clinical sample, it’s unclear what this implies. The study limitations should also be more than the small sample size alone.
Author Response
Comment 1. This study investigates how the COVID-19 may have affected individuals on anxiety, worry and trait mindfulness. The findings have potentials to contribute to the current knowledge on coping with the current pandemic and their trait may play a role in such disruptive and distressful events. The study has apparent benefits, and yet the study design (or its presentation) could be much improved. With the extremely small same size, first of all, this study should include more descriptions on the sample and further elaborate its limitations based on them. No confounding factors were considered in the study, which may critically limit the study’s interpretation and its usage. Given such small sample size, it could be much stronger with qualitative information of the samples and other external factors (other than traits). A mixed-method approach would make this study more meaningful and informative than as is. Given worry/anxiety is intricate and often conflated with situational (state) components, I strongly recommend adding rich qualitative data to consider and inform about confounding factors for the key variables.
Response. We fully agree with the Reviewer that with such a small sample size the study should have included more descriptions on the sample in order to include qualitative information and likely consider mixed methods integrating quantitative and qualitative data. However, recruitment of the participants for the study was the result of a fortunate occasion, since all the subjects completed the self-report measures at pre-lockdown assessment as part of the research activities at our Laboratory and, then, most of them gave their willingness to complete again, at T1, the same self-report measure as before. In this context, no further qualitative information was collected other than sex, age and education. Thus, we are not able to provide the qualitative data that could have better clarified our results. In the revised manuscript, we commented on this point within the study limitations. However, notwithstanding this limitation the study design allowed us to test a specific hypothesis, namely whether individual’s proneness to worry could affect emotional changes during the quarantine and whether trait mindfulness could be related to worry and anxiety, working as a sort of protecting factor to emotional distress. Our results revealed that individuals with high trait worry at pre-lockdown showed at the end of lockdown a significant increase of anxiety sensitivity and, in particular, of concerns about own mental health with respect to low worriers. Moreover, following the Reviewer suggestion (Comment 2), we also found in the whole sample that trait mindfulness was inversely related with worry and fear of mental health. Together, these findings provide valuable hint for tailoring psychological interventions for at-risk individuals enhancing their ability to focus attention and awareness on the present moment through mindfulness-based interventions (please also see Comment 3).
Comment 2. The study could also be looked at from MAAS score high vs. low (e.g. trait worry presented in Table 2 in reverse) and its (trait mindfulness’s) relation with trait worry should be further clarified in text.
Response. We thank the Reviewer for the suggestion that allowed to better clarify the role of trait mindfulness in the present study. In the revised manuscript, a two-tailed Mann-Whitney U test was used to assess differences between high and low trait mindfulness in the two conditions (T0 vs T1), but no significant differences on PSWQ, ASI-3, and MAAS were found (we did not provide this analysis in the text but we could add it if suggested). Instead, we provided in the text the results of the Spearman’s correlations between the measures, separately at T0 and T1. These results revealed that, at T0, the MAAS score showed a significant negative correlation with both ASI-3 total score (rho= -.491; p= .013) and ASI-3 cognitive concern score (rho= -.573; p=.003). At T1, correlational analysis revealed a significant negative correlation between the MAAS score and both ASI-3 cognitive concern score (rho= -.405; p=.044) and PSWQ score (rho= -.505; p= .010). We discussed these last findings suggesting a possible role of trait mindfulness in protecting people from maladaptive worrying during the COVID-19 lockdown. Moreover, we suggested that interventions enhancing the individual disposition to mindfulness through specific mindfulness-based training programmes are warranted (see Comments 1 and 3).
Comment 3. Another critical issue that authors would want to address is how this study may actually contribute to any potential practice or policy interventions. The (probably only) implication authors made was on p. 5 of 7. L. 184-6. “effort in detection early signs of clinical mental health problems is of primary relevance to deal with the psychological crisis due to COVID‐19 pandemic, especially in at-risk populations.” Since the study results are from non-clinical sample, it’s unclear what this implies. The study limitations should also be more than the small sample size alone.
Response. We thank the Reviewer for having underlined this point. As reported in responding to the previous comments, following results of the correlation analyses showing that trait mindfulness was inversely related to both worry and fear of mental health, we could provide in the revised text a more precise suggestion about a possible useful intervention to counteract the negative effects of worry on fear of psychological dyscontrol. Indeed, we suggested that a viable strategy for supporting at risk individuals could be to enhance their ability to focus attention and awareness on the present moment through mindfulness-based training interventions.
Reviewer 2 Report
This paper presents an analysis of some interesting data from an unintended natural experiment, comparing change in a number of anxiety-related outcomes before and after pandemic lockdown among participants who had higher and lower levels of trait anxiety at pre-pandemic baseline. Although the sample size is relatively small, this study addresses a very timely issue with a good design. There are a few points that could be clarified to improve the manuscript.
- It seems unexpected that 60% of the sample would score 1.5 SD or more above the population mean for trait worrying. Were the participants screened prior to enrollment to include a sufficient number with high trait worrying, or did this happen by chance? Either way, this should be briefly discussed in the methods section.
- In the first paragraph on page 5 it is stated that high and low worriers did not differ in terms of the physical and social anxiety scales, but it does not appear that this is accurate, since the high worrying group means for both of these variable are roughly 2 SD higher than the low worrying group at T1. Instead, it would be accurate to say that there was no significant mean change between T0 and T1 in either the high or low worrying group.
- It looks like there are some decimals that have been omitted in Table 2 (e.g., the low worriers mean at T1 should presumably be 7.60 rather than 760).
Author Response
Comments and Suggestions for Authors
This paper presents an analysis of some interesting data from an unintended natural experiment, comparing change in a number of anxiety-related outcomes before and after pandemic lockdown among participants who had higher and lower levels of trait anxiety at pre-pandemic baseline. Although the sample size is relatively small, this study addresses a very timely issue with a good design. There are a few points that could be clarified to improve the manuscript.
Response. We thank the Referee for the positive comment.
Comment 1. It seems unexpected that 60% of the sample would score 1.5 SD or more above the population mean for trait worrying. Were the participants screened prior to enrollment to include a sufficient number with high trait worrying, or did this happen by chance? Either way, this should be briefly discussed in the methods section.
Response. The participants were not screened prior to enrollment, and this high percentage of high worry participants happened by chance. However, we did not report in the original version of the text that our original sample was of 31 participants, and that 6 out of 31 refused to participate to the second assessment at T1. In the revised manuscript we have specified this point, although we are not able to establish whether this aspect could have contributed to define the features our final sample. Thus, we are not in the position to clarify the reason why such a high percentage of participants showed high worry. However, it is important to underscore here that our percentages were consistent with the classical view that both clinical and subclinical anxiety and anxiety-related problems are higher in university students than in general population (e.g., Connell and Barkham 2007; Cooke et al. 2006; Eisenberg et al. 2011). This point has been discussed in the revised text as a study limitation on the generalizability of our results.
Comment 2. In the first paragraph on page 5 it is stated that high and low worriers did not differ in terms of the physical and social anxiety scales, but it does not appear that this is accurate, since the high worrying group means for both of these variable are roughly 2 SD higher than the low worrying group at T1. Instead, it would be accurate to say that there was no significant mean change between T0 and T1 in either the high or low worrying group.
Response. We thank the Reviewer for this comment. In the revised manuscript we modified the accordingly.
Comment 3. It looks like there are some decimals that have been omitted in Table 2 (e.g., the low worriers mean at T1 should presumably be 7.60 rather than 760).
Response. We thank the Reviewer for having underscored the typo in Table 2. We amended it.
Reviewer 3 Report
This is a timely piece given the current COVID19 pandemic. The authors were indeed in a unique position to look at mental health and COVID19 in a group of young adults. Moreover, the opportunity presented to investigate trait mindfulness as a protective factor and sex differences is novel.
Line 40 - remove "so called"
Line 39 - from "Moreover..." - split the sentence into two separate sentences. It's too long and doesn't read well.
Line 44-45 - explain "healthy participants" in the context of the Chinese study
Line 46: remove "As regard", replace with "In"; also add COVID19 to the sentence so it is clear that the dysfunctional personality problems are related to a study done during COVID19.
Line 48: replace "recent" with "2020"
Line 50 - 51: How are duration of quarantine and inadequate information a psychological consequence of quarantine? How did the meta-analysis frame the criteria of inclusion for psychological consequence? I can understand that the duration of quarantine may lead to worry or anxiety, but duration of quarantine itself is not a psychological consequence.
Line 66: replace "contrariwise" with "conversely"
Line 67: check in text citation 16 - it isn't in brackets
Line 72: replace "(university students)" with young adults attending University
Line 76: Do you mean "in comparison" rather than "with respects"
Line 84: This is a very small sample. Is it that only 25 participants had an outcome of "high worry" as per the inclusion criteria? If so, it must be made explicit.
Line 92: Which online platform did you use for questionnaire administration?
Line 95-97: While I do understand that the procedures followed during data collection were in accordance with the Helsinki Declaration was this component approved by a University ethics research committee or board? Or was an amendment granted on the initial study to extend the study to during lockdown? Also, was consent embedded in the online questionnaire or was this separate done with those who agreed to participate.
Line 135: what about including the explanation of trait mindfulness in the results and its role as a protective factor?
Line 192: Another limitation is that the study was limited to young adult university students. Thus, in addition to the need for a larger sample size, there is also a requirement for a broader representation of the population.
Line 202-203: Expand a bit more on the tailored interventions by giving an example of what this would be. In its current form, the recommendation for intervention is very broad.
Author Response
Comments and Suggestions for Authors
This is a timely piece given the current COVID19 pandemic. The authors were indeed in a unique position to look at mental health and COVID19 in a group of young adults. Moreover, the opportunity presented to investigate trait mindfulness as a protective factor and sex differences is novel.
Response. We thank the Reviewer for the positive comment.
Comment 1. Line 40 - remove "so called"
Response. Thanks. Done.
Comment 2. Line 39 - from "Moreover..." - split the sentence into two separate sentences. It's too long and doesn't read well.
Response. Thanks. Done.
Comment 3. Line 44-45 - explain "healthy participants" in the context of the Chinese study.
Response. Thanks for having underscored this point. We modified the text specifying that participants of the Chinese study were University students.
Comment 4. Line 46: remove "As regard", replace with "In"; also add COVID19 to the sentence so it is clear that the dysfunctional personality problems are related to a study done during COVID19.
Response. We thank the Reviewer for the suggestion. We modified the text accordingly.
Comment 5. Line 48: replace "recent" with "2020"
Response. Thanks. Done.
Comment 6. Line 50 - 51: How are duration of quarantine and inadequate information a psychological consequence of quarantine? How did the meta-analysis frame the criteria of inclusion for psychological consequence? I can understand that the duration of quarantine may lead to worry or anxiety, but duration of quarantine itself is not a psychological consequence.
Response. In the revised version of the manuscript we rephrased the sentence as follow: “In a 2020 meta-analysis, the main psychological consequences of quarantine during the 2003 SARS and the 2014 Ebola were described with fears of infection, frustration and boredom. Also, the duration of quarantine and inadequate information contributed to psychological distress”.
Comment 7. Line 66: replace "contrariwise" with "conversely"
Response. Thanks. Done.
Comment 8. Line 67: check in text citation 16 - it isn't in brackets
Response. Thanks. Done.
Comment 9. Line 72: replace "(university students)" with young adults attending University
Response. Thanks. Done.
Comment 10. Line 76: Do you mean "in comparison" rather than "with respects"
Response. We modified the text as follows: “We expected that participants with higher levels of pre-existing worry could display stronger anxiety symptoms during quarantine when compared to individuals with low levels of worry”.
Comment 11. Line 84: This is a very small sample. Is it that only 25 participants had an outcome of "high worry" as per the inclusion criteria? If so, it must be made explicit.
Response. As reported in response to Comment 1, the participants were not screened prior to enrolment, and this high percentage of high worry participants happened by chance. However, in the revised text we specified that our original sample was of 31 participants, and that 6 out of 31 refused to participate to the second assessment at T1.
Comment 12. Line 92: Which online platform did you use for questionnaire administration?
Response. In the revised version of the text we specified that Google Forms online platform was used.
Comment 13. Line 95-97: While I do understand that the procedures followed during data collection were in accordance with the Helsinki Declaration was this component approved by a University ethics research committee or board? Or was an amendment granted on the initial study to extend the study to during lockdown? Also, was consent embedded in the online questionnaire or was this separate done with those who agreed to participate.
Response. The research protocol was approved by the Local Ethics Committee and, as correctly suggested by the Reviewer, the assessment at the second time point was an amendment granted on the initial approval of the research. Moreover, the informed consent form was separate from the questionnaires and it was provided only to persons who agreed to participate. Thus, persons willing to participate provided their formal consent and then they were allowed to complete the questionnaires.
Comment 14. Line 135: what about including the explanation of trait mindfulness in the results and its role as a protective factor?
Response. We thank the Reviewer for the suggestion. In the revised version of the text, we added some possible explanations on the role of trait mindfulness in the anxiety response during the COVID-19 lockdown. Moreover, we took into account this point when discussing hints for implementing psychological interventions to help at risk persons to tackle worry and anxiety.
Comment 15. Line 192: Another limitation is that the study was limited to young adult university students. Thus, in addition to the need for a larger sample size, there is also a requirement for a broader representation of the population.
Response. We thank the Reviewer for this suggestion. In the revised text we added this issue as a further limitation.
Comment 16. Line 202-203: Expand a bit more on the tailored interventions by giving an example of what this would be. In its current form, the recommendation for intervention is very broad.
Response. As underscored in response to Comment 14, in the revised text (Conclusion section) we could be more precise about recommendations for intervention. Indeed, A following results of a new correlation analyses showing that trait mindfulness was inversely related to both worry and fear of mental health, we could provide in the revised text a more suggestion about a possible useful intervention to counteract the negative effects of worry on fear of psychological dyscontrol. In particular, we suggested that a viable strategy for supporting at risk individuals could be to enhance their ability to focus attention and awareness on the present moment through mindfulness-based training interventions.